# Right Place at the Right Time: How Changes in Protocadherins Affect Synaptic Connections Contributing to the Etiology of Neurodevelopmental Disorders

**DOI:** 10.3390/cells9122711

**Published:** 2020-12-18

**Authors:** Maria Mancini, Silvia Bassani, Maria Passafaro

**Affiliations:** 1Consiglio Nazionale delle Ricerche (CNR), Istituto di Neuroscienze, 20129 Milan, Italy; maria.mancini@in.cnr.it (M.M.); silvia.bassani@in.cnr.it (S.B.); 2NeuroMi, Milan Center for Neuroscience, 20126 Milan, Italy

**Keywords:** protocadherins, neurodevelopmental disorders, cell adhesion molecules, synaptic protein

## Abstract

During brain development, neurons need to form the correct connections with one another in order to give rise to a functional neuronal circuitry. Mistakes during this process, leading to the formation of improper neuronal connectivity, can result in a number of brain abnormalities and impairments collectively referred to as neurodevelopmental disorders. Cell adhesion molecules (CAMs), present on the cell surface, take part in the neurodevelopmental process regulating migration and recognition of specific cells to form functional neuronal assemblies. Among CAMs, the members of the protocadherin (PCDH) group stand out because they are involved in cell adhesion, neurite initiation and outgrowth, axon pathfinding and fasciculation, and synapse formation and stabilization. Given the critical role of these macromolecules in the major neurodevelopmental processes, it is not surprising that clinical and basic research in the past two decades has identified several *PCDH* genes as responsible for a large fraction of neurodevelopmental disorders. In the present article, we review these findings with a focus on the non-clustered PCDH sub-group, discussing the proteins implicated in the main neurodevelopmental disorders.

## 1. Introduction

Neurodevelopmental disorders are a group of diseases occurring in early life and characterized by a significant alteration of the central nervous system (CNS) functioning, resulting in the failure to meet the typical developmental milestones. Brain dysfunction can manifest in several different ways ranging from intellectual alterations and problems of learning and communication to motor function impairment. Although the different disorders have a heterogeneous etiology, there is often overlap of symptomatology between the numerous neurodevelopmental disorders [1,2], and association with many co-morbidities including epilepsy, mood disorders, impairment in vision and hearing, sleep disturbances, and gastrointestinal and breathing problems. Many studies have suggested that shared molecular pathways could account for the multiple clinical signs that characterize these diseases [3]. The altered sociability, often observed in individuals with neurodevelopmental disorders, makes such diseases a significant and expanding public health problem. The therapeutic options available to treat the symptoms are limited. Thus, there is a great need to understand the core neuropathology and the underlying molecular mechanisms to identify new therapies.

Many neurodevelopmental disorders are thought to result from an interaction between genetic and environmental factors [4]. However, in some cases, the disorders can be traced to genetic abnormalities ranging from single nucleotide changes to loss or gain of up to thousands of nucleotides and chromosomal rearrangements. A growing body of literature suggests that mutations or deficits in genes that regulate synapse and circuit development and/or function lead to neurodevelopmental disorders [5]. Regarding the genes encoding synaptic proteins, the prevalence of mutations has been found in the synaptic cell adhesion molecules (CAMs), including protocadherins (*PCDH*s). In this review, we will summarize the current knowledge about PCDHs, a group of proteins regulating multiple aspects of the synapse development, function and plasticity. We will focus on selected non-clustered PCDHs discussing their implication in neurodevelopmental disorders.

## 2. Protocadherins in the Central Nervous System

### 2.1. Overall Structure and Classification

PCDHs are single membrane-spanning glycoproteins and constitute the largest sub-group of the cadherin superfamily, cell-surface molecules implicated in the regulation of the cell-cell contacts.

Cadherins consist of an extracellular region subdivided into extracellular cadherins (EC) domains, each including a repetitive sequence named cadherin motif, and a cytoplasmic domain (Figure 1) [6]. The cadherin motif contains conserved Ca^2+^-binding sequences that are required for protein functioning. The cytoplasmic domain interacts with the armadillo repeat proteins p120 catenin and β-catenin [7,8,9], forming the cadherin–catenin complex that is crucial for the mechanical adhesion between cells. Based on shared properties and sequence similarity cadherins are organized in sub-groups: the classical type I and the related type II, desmosomal cadherins, and PCDHs (Figure 1).

In humans, the cadherin superfamily includes over 100 members, more than half of which are PCDHs [10]. Although PCDHs are very similar to the classical cadherins for the presence of a single transmembrane domain and the conserved cadherin motifs, they present some differences. Unlike cadherins, which present 5 EC repeats, PCDHs have 6 or 7 EC repeats (in few cases more) that show low sequence similarities to the EC domains of the classical cadherins (Figure 1). Differences in the cytoplasmic domain, where the catenin-binding sites result absent, have also been highlighted from the functional point of view.

PCDHs are encoded by more than 70 different genes (*PCDH* genes). According to the genomic organization, PCDHs are divided into clustered and non-clustered PCDHs (Figure 1) [11]. Two additional sub-groups, known as atypical PCDHs, have been identified and include the seven-transmembrane PCDHs and the giant Fat PCDHs.

In clustered PCDHs, which include the three families PCDHα, PCDHβ, and PCDHγ, the gene clusters are arranged in tandem in a small genome locus on a single chromosome (human chromosome 5q31; mouse chromosome 18) [12]. Every gene cluster presents several variable exons, each involved in the generation of an extracellular domain, a transmembrane domain, and a variable portion of the cytoplasmic domain. Downstream to the variable exons, only in *PCDHα* and *γ* cluster genes, it is possible to find three constant exons coding for a shared portion of the cytoplasmic domain (Figure 2). The presence of variable exons and this gene organization allows the production of more than 50 PCDH isoforms from the three gene clusters through alternative splicing or through the choice of alternative promoters.

Conversely, non-clustered PCDHs do not present a clustered genome locus but the genes are dispersed on multiple chromosomes. Although the genes in non-clustered PCDHs produce alternative splicing variants, they do not encode variable extracellular domains; this results in the possibility to generate only small variations in the protein product.

Non-clustered PCDHs are organized into two sub-groups: PCDHδ and solitary PCDHs, also known as PCDHε [11]. On the basis of the homology, the number of EC repeats, and the amino acid sequence of the cytoplasmic domain, PCDHδ family can be further divided into two sub-groups, PCDHδ1, and PCDHδ2. While PCDHδ1 has 7 EC domains and 3 conserved cytoplasmic motifs, CM1, CM2, and CM3, PCDHδ2 has 6 EC domains and only 2 cytoplasmic motifs, CM1 and CM2 (Figure 1 and Figure 3). δ1 sub-group includes PCDH-1, 7, 9 and 11; δ2 sub-group contains PCDH-8, 10, 17, 18 and 19. In addition to the nine canonical family members, in δ2 and δ1 subfamily there are PCDH12 and PCDH20, respectively, that diverge in their intracellular regions characterized by the absence of the CM1, CM2, and CM3 motifs (Figure 3).

PCDHε sub-group includes PCDH-15, 16, 21, and MUCDHL (Figure 3). Except PCDH21, the other PCDHs in this sub-group are characterized by the presence of higher or lower number of EC repeats compared to PCDHδ; in fact, PCDH15, PCDH16, and MUCDHL have 11, 27 and 4 cadherin repeats, respectively.

The two families of large atypical PCDHs include those typified by Drosophila Fat, a giant protein with 34 EC repeats in its extracellular domain, and those typified by Drosophila Flamingo, a 7-transmembrane (7TM) domain protein with 8 EC repeats, along with their many vertebrate homologues [13].

### 2.2. Tissue Distribution

PCDHs are primarily expressed in the CNS and regulate multiple aspects of cell interaction, synapse maturation, synapse function and plasticity, essential for the proper brain development and correct functioning. Like classical cadherins, PCDH expression is spatiotemporally regulated during brain development in several vertebrate species. PCDHs are constitutively distributed in the cortex, hippocampus, amygdala, cerebellum, thalamus, hypothalamus, basal ganglia nuclei, limbic system structures, olfactory system, and visual system (Table 1) [14,15], but some of them show prominent gradients or regionalized expression [15,16,17] reflecting a functional differentiation. Moreover, due to the projections to different functionally related areas, for example from the cortex to the striatum, the presence of selective PCDHs strongly impacts the activity of the whole circuitry.

## 3. Functional Roles of Neuronal Protocadherins

### 3.1. Cell-Cell Adhesion

Like cadherins, PCDHs are synaptic calcium-dependent adhesion glycoproteins that take part in the process of cell aggregation and regulate cell–cell interactions. They have been detected in synaptic and extrasynaptic membranes as well as in the intracellular compartments [12,18,19]. Cells can vary the number of PCDHs expressed, the level of surface expression, and the kind of PCDHs exposed. The combination of expression of different PCDHs on the neuronal surface seems to strongly influence the cell functions, as different members possess distinct apparent adhesive affinities [20].

The interaction between pre- and post-synaptic sites occurs through connections of the extracellular domains of PCDH isoforms. The engagement of PCDH isoform-specific dimers is mediated by the EC domains. For both the clustered and non-clustered PCDHs of the δ family, the EC1-EC4 domains, combined in an antiparallel orientation (head-to-tail), mediate the formation of the *trans* (cell-cell) dimers (Figure 4) [21,22]. *Cis* (same-cell) interactions have also been shown in clustered PCDHs (Figure 4) [23] and are nonselective between different isoforms. This aspect further contributes to create an enormous combinatorial diversity. Whether PCDHs of the δ family are involved in similar *cis* interactions is not clear yet, although a recent investigation seems to exclude such possibility [24].

Non-clustered PCDH-1, 7, 8, 10, 18, and 19 exhibit homophilic trans binding activity that is preferred to the heterophilic interactions [25]. However, for some of these molecules (PCDH-8, 10, and 19), the strength of the homophilic binding is weak [26,27,28]. In the case of PCDH8, any cell aggregation has been observed; by contrast, PCDH8 has been shown to interfere with the classic cadherin-mediated adhesion [29,30] that, however, can be restored when the cytoplasmic tail of PCDH8 is removed [31]. Similarly, in the case of PCDH19, the strength of the cell-cell adhesion can be enhanced after removal of the cytoplasmic domain, suggesting that this region may not stabilize the interaction to facilitate adhesion. Conversely, it negatively regulates the adhesion mediated by the extracellular domain [28].

Although less frequent, heterophilic interactions have been reported between PCDH15 and cadherin 23 [32], PCDHγ-C5 and GABA(A) receptor (GABA(A)R [33] and between the clustered PCDHα4 and β1 integrin [34]. This latter is due to the presence of the RGD motif, a peptide sequence of three amino acids (Arg-Gly-Asp) that is recognized by integrins. The RGD motif has been seen not only in the extracellular domain of clustered, but also non-clustered PCDHs (PCDH-17, 19 and MUCDHL), raising the intriguing possibility that such proteins may be ligands for integrins through heterophilic bindings.

### 3.2. Synapse Maturation and Circuit Formation

The neuronal specificity of wiring in the CNS is achieved at different levels during the development. In large part, this is due to the action of CAMs, including PCDHs, and recognition proteins, which allow cells to sense the environment and establish selective interactions between the enormous number of other neurons and glial cells. It has been proposed that the combination of PCDHs expressed on the neuronal surface generates a recognition unit [23,25,35] that prevents the interaction at synapses expressing matching PCDHs (self-avoidance) [36,37], or facilitates the adhesion between synapses from different cells (Figure 5A). This makes PCDHs crucial during neurodevelopment and implicates them in both synaptic destabilization/pruning and synaptic stabilization and maturation.

PCDHs participate both in early and in late neurodevelopmental events. During the embryonic stages, PCDHs finely process the dendritic arbors of the post-mitotic neurons in the correct spatial orientation and control the axons outgrowth along the right paths (Figure 5B) [38,39,40,41,42,43].

The importance of PCDHs in dendritic arbor formation has been demonstrated by Garrett and colleagues [44] in cortically restricted PCDHγ mutant mice. Analogously, Suo, and colleagues [45] have found that PCDHα mutant hippocampal neurons have simplified dendritic arbors and a reduction in dendritic spine density. A similar phenotype has been observed in serotoninergic neurons of the rostral raphe nuclei from the same mutant mice [42]. Together, these findings provide evidence that both PCDHγ and α proteins promote dendrite arborization.

Defects in PCDHs, have also been correlated to defects in the formation and the extension of axonal tracts [46,47]. For example, while the reduction of PCDH10 seems responsible for defects in the outgrowth of the striatal axons [46,48], the absence of PCDH17 has been considered determinant in causing defects in the extension of axons from specific subdivisions of the amygdala. In vitro experiments of migration, performed using amygdala neurons from *PCDH17* KO mice, have in fact shown that the growth cones of the axons stop moving at the contact points with other encountered axons instead of cross them [47], clearly supporting the hypothesis that PCDH17 serves to aid the axonal sorting. Other PCDHs involved in axon outgrowth and guidance are PCDH7 and PCDH18b, as shown following interference experiments [49,50,51].

Interestingly, also PCDHγ proteins, in addition to the dendritic arbor formation, have been implicated in the correct axonal development. Although studies on PCDHγ mutant mice show normal axonal outgrowth, targeting, and formation [44,52], alterations in the patterning of the axonal terminals have been observed in PCDHγ mutant mice. In mice where all 22 *PCDHγ* genes were deleted, it has been observed a loss of synapses in the late embryonic period with a reduction of the excitatory and inhibitory synaptic puncta by 40–50% [52] that has been ascribed to the absence of ubiquitiously-expressed three C-type variable exons C3, C4 and C5 rather than to the absence of the sparsely-expressed A1, A2 and A3 exons of the clustered PCDHs [53].

As neurodevelopment proceeds, PCDHs play a crucial role in the interaction between axons and dendrites to regulate the formation of excitatory synapses (Figure 5C) [54]. The presence of PCDHs at synaptic level has been confirmed in synaptosome preparations and post-synaptic density protein fractions [55,56,57]. Similarly, immunohistochemistry and electron microscopy experiments have confirmed PCDHs distribution at some, but not all, synapses; however, these technical approaches, have highlighted the presence of much of the proteins at perisynaptic level or in other compartments of dendrites and axons [56]. The differential expression of PCDHs and their compartmental distribution define the number of synapses generated in a given region. In some cases, the formation of new synapses is favored, whereas, in other cases, there is a downregulation in the number of synapses. For example, *PCDH-γ*s KO mice show increased synapses in cortical neurons [58] but a reduced number of synapses in spinal cord neurons [52]. Furthermore, the number of synapses has been found to be decreased not only in vivo but also in cultured hippocampal neurons where *PCDH-γ*s had been knocked down using shRNA [45]. These differences seem to be mainly correlated to the cell type.

Changes in the number of the dendritic spines have been observed also in arcadlin/PCDH8 KO mice [59] and in *PCDH10*-heterozygous mice [60]. In both cases, the absence or even the reduction of these proteins determines an increase in the spines available, suggesting that these molecules are involved in the elimination of inappropriate or unused synapses during the normal development process.

In addition to the formation of new spines, PCDHs seem to be implicated also in the progressive accumulation of synaptic vesicles at the presynaptic level, another distinctive feature of the synaptic maturation (Figure 5C). A study performed on *PCDH17* KO mice has reported an enhanced presynaptic vesicle accumulation in both excitatory and inhibitory synapses of corticobasal ganglia [61]. The investigation of the functional synaptic maturation in the absence of PCDH17 has demonstrated altered synaptic transmission efficacy that has an impact on the circuit formation and activity.

The study on PCDH17 by Hoshina and colleagues [61], together with investigations on PCDH10 and other non-clustered PCDHs [14,26,62], have highlighted the importance of PCDHs in determining a precise functional topography in the brain and a pathway-specific synapse development necessary for the neural circuit formation. Only neurons of a given functional network present the same repertoire of PCDHs. For example, the expression of PCDH17 in the cortex, basal ganglia, and thalamus [61], is opposite to that of PCDH10 [14,26,62]. Moreover, the expression pattern can be distinctive of a specific phase of the neurodevelopment. This makes PCDHs rarely overlapping in a specific brain region; conversely, they complement each other in their functions and in the proper circuit development.

## 4. Protocadherins in Neurodevelopmental Disorders

Given the wide distribution of PCDHs in the vertebrate CNS, and the fact that they are involved not only in the developmental steps but also in synapse maturation and functioning, as well as in circuit formation, it is perhaps not wholly surprising that this class of molecules is involved in a large number of CNS disorders.

To date, PCDHs have been implicated in several neurodevelopmental disorders such as intellectual disability (ID), epilepsy, Fragile-X syndrome (FXS), and Autism spectrum disorders (ASDs) and have been demonstrated to fulfill roles in disease states conceptualized as neurodevelopmental disorders including schizophrenia (SZ) and bipolar disorder (BD) (Table 2).

Although certain *PCDH* genes have been assigned as major genes causative of a specific neurodevelopmental disorder, the advancement in genome-wide analysis studies have identified many *PCDH*s genes as susceptibility genes or risk factors for a particular disorder. Below we have summarized recent progress in the study of some non-clustered PCDHs that have been implicated in several neurodevelopmental diseases.

### 4.1. PCDH19

PCDH19, a member of the PCDHδ2 sub-group, has attracted the attention of the scientific community since 2008. This year represented a turned point for PCDH19 research, as *PCDH19* was recognized as the gene responsible for the neurodevelopmental syndrome known as Developmental and Epileptic Encephalopathy 9; DEE9 (OMIM # 300088) [63]. This syndrome, also known as Epilepsy, Female-restricted, with Mental Retardation (EFMR) [64], was first reported in 1971 as a convulsive disorder limited to females [65] (Table 2).

DEE9 peculiar feature is the recurrence of seizures clusters with childhood onset and commonly observed remission during adolescence, often resistant to drug treatments. Frequent comorbid symptoms are ID of variable degree and a wide spectrum of behavioral and psychiatric symptoms, ranging from ASDs to schizophrenia [63,65,66,67,68].

More than 175 mutations have been reported so far in *PCDH19* [69]. A significant mutations proportion consists of missense point mutations affecting the PCDH extracellular domain. However, alternative mutation types (i.e., nonsense point mutations, small insertions/deletions, whole-gene deletion) and locations (i.e., the intracellular domain) have been reported [69,70]. Since no clear correlation has been observed between the different mutations and phenotype severity, pathogenic variants might ultimately lead to protein loss of function [67,69].

Notably, *PCDH19* mutations cause DEE9 only when expressed in heterozygosis. *PCDH19* is located on the chromosome X (Xq22) and females with a mutant allele and a wild type allele develop DEE9, while hemizygous males with a unique mutant allele do not. In females, one of the two X chromosomes randomly undergoes inactivation, which implies that cells expressing a functional and a mutated *PCDH19* allele coexist in DEE9 female patients. It has been hypothesized that this mosaic condition might cause cellular interference [71]. How these two cell populations, which diverge in PCDH19 expression, would interfere with each other in practice remains unknown. We can speculate that this condition might scramble cell-cell communication. In fact, the specific set of adhesion molecules that each cell expresses on its surface defines its identity. Adhesion molecules allow cells to recognize themselves and trigger bidirectional intracellular signaling cascades that ultimately affect the assembly of neuronal circuits. An altered expression of PCDH19 might confer fake identities to neurons and convey altered messages that might affect circuit wiring. In support of the cellular interference hypothesis came the DEE9 diagnosis for some male patients with a mosaic expression of PCDH19, due to a post-zygotic somatic mutation in their unique *PCDH19* allele [68,71,72,73,74,75,76,77] or to the mutation in one of the two *PCDH19* alleles in the context of Klinefelter syndrome (XXY) [78].

PCDH19 expression in rodent brain extends from the embryonic development [14,79] to adulthood [17,80]. In the cortex and hippocampus, PCDH19 expression peaks in the first postnatal period [81,82], suggesting PCDH19 involvement in brain maturation. In the first postnatal week, PCDH19 is highly expressed in the limbic structures [80,83] where patients’ seizures originate [84], including the amygdala, hippocampal regions CA1-CA3, and subiculum, and some hypothalamic areas. While PCDH19 is not detectable in dentate gyrus (DG) hippocampal neurons at this stage, its expression becomes evident in adulthood [17,80]. In addition to neurons, PCDH19 expression was observed also in endothelial cells [80], which is consistent with a putative role of PCDH19 in maintaining the blood-brain barrier integrity [85].

During the embryonic period, PCDH19 is highly expressed in neuronal progenitors, and an increasing body of evidence highlights a role of PCDH19 in regulating progenitor cells polarity, proliferation and fate [82,86,87,88]. In particular, PCDH19 expression was shown to be important for the proliferation of radial glia, from which all cerebral cortex neurons originate [82]. In a recent paper, Xiaohui et al. demonstrated that PCDH19 downregulation in mouse brain decreased the number of excitatory neurons, but not glial cells, arising from individual radial glial progenitors [88], thus supporting the role of PCDH19 in neurogenesis during cortical embryonic development. PCDH19 expression was also important to preserve the physiological lateral clustering of clonally related excitatory neurons (i.e., neurons arising from a common progenitor) and their preferential intra-clone synaptic connections [88]. These and additional findings indicate that PCDH19 role goes beyond the regulation of neuronal number and includes neuronal spatial distribution, maturation, and connectivity. Altered sorting pattern of neuroprogenitors was observed in heterozygous knock-out (KO) mice with PCDH19 mosaic expression [89,90]. Abnormal migration and morphological maturation were observed in rat hippocampal neurons in which PCDH19 expression was downregulated [81]. In Zebrafish, pcdh19 loss impaired the columnar organization of optic tectum neurons [91] and affected the developmental trajectories of neuronal network functional properties [92].

Cell-cell recognition mechanisms during migration and circuit formation likely rely on PCDH19 adhesive properties, both *in trans* with itself and *in cis* with *N*-cadherin [93,94,95]. Furthermore, PCDH19 has been shown to associate with both the WASP-family verprolin homologous protein (WAVE) regulatory complex, which controls actin cytoskeleton dynamics, and with regulators of Rho GTPases and of the microtubule cytoskeleton [94,96,97] and this likely explains its role in cell proliferation and neuronal maturation.

PCDH19 role in neuronal proliferation, migration, and maturation provides an explanation for the structural defects recently described in DEE9 patients’ brain. Radiological analyses revealed focal cortical malformations [98] and quantitative magnetic resonance imaging (MRI) allowed in-depth study of morphological abnormalities in the limbic cortex. The cortical surface morphology and the underlying white-matter bundles were altered in DEE9 patients, as indicated by local gyrification reduction and bundles abnormal directional diffusion, respectively. These defects are the likely structural counterpart of functional connectivity defects [99].

Mouse models were able to recapitulate DEE9 core symptoms, such as cognitive and social behavior impairment and seizure susceptibility. Hippocampal and amygdala-dependent deficits have been observed in heterozygous female KO mice but not in hemizygous KO males tested in the fear conditioning paradigm [89], in agreement with the cellular interference hypothesis. By contrast, both KO groups, i.e., heterozygous females and hemizygous males, showed autism-like behaviors, evaluated as reduced sociality in the 3-chamber test [100]. This is consistent with inflexible personality and obsessive traits observed in some male patients with PCDH19 mutations [67]. Even though spontaneous epilepsy has not been reported, increased seizure susceptibility has been observed both in rats, in which PCDH19 had been downregulated in a subset of hippocampal neurons [81], and in *PCDH19* KO mice [101]. Surprisingly, both homozygous and heterozygous KO female mice displayed enhanced susceptibility to induced seizures, contrary to KO males [101]. These data challenge the cellular interference hypothesis and suggest that (i) PCDH19 expression is not totally dispensable, since its complete loss can associate with autistic traits and (ii) other determinants such as gender, in addition to PCDH19 mosaic expression, might be at play.

With this regard, it has been observed that seizures in DEE9 female patients occur in a time-window in which sex hormones are low. Furthermore, DEE9 female patients display altered expression of neurosteroid hormones metabolizing enzymes in their skin fibroblasts and reduced level of allopregnanolone in blood [102]. Allopregnanolone is a potent GABA(A)R modulator [103]. Notably, PCDH19 has been shown to interact with the GABA(A)R and to modulate its surface expression and kinetics in hippocampal neurons, with important consequences for both tonic and phasic inhibition and ultimately neuronal excitability [81,104]. The clinical efficacy of an allopregnanolone analog, ganaxolone, is currently under evaluation [105,106] and represents a promising therapeutic option to face DEE9-related epilepsy, for which current treatments are unsatisfactory.

### 4.2. PCDH10

PCDH10, also known as OL-protocadherin, is a member of the PCDHδ2 sub-group with a unique cytoplasmic domain. This protein, endowed with homophilic adhesion activity, mediates target recognition, and synapse formation [48]. It is largely expressed in several brain regions during development. Certain local circuits, including the olfactory and visual system, as well as the olivocerebellar projections, are particularly enriched in PCDH10 [26,62,107]; however, the cerebral districts where PCDH10 is very abundant are the basal ganglia system and the amygdala.

The studies on PCDH10 in the basal ganglia have stressed its role of regulator of the axonal growth and of the synaptic connection development, as well as its implication in cognition, motor activity and emotion, three behavioral domains supported by the corticostriatal-thalamic circuits. Defects at the cellular level and abnormal connectivity in the subcortical structures (caudate, dorsal striatum, and thalamus) are considered responsible for the atypical development of these networks and for the onset of repetitive behaviors, cognitive inflexibility, and abnormalities in the motivated behaviors observed in ASD [108] (Table 2). At a striatal level, PCDH10 has been found essential for the outgrowth of the striatal axons that, in turn, is necessary for the formation of the thalamocortical projections [46,48]. The process of elongation of the striatal axons has been hypothesized to be correlated to the ability of PCDH10 to modify the cytoskeleton dynamics and promote the rearrangement of actin filaments that, destabilizing the cell adhesion, could promote the axonal outgrowth [109]. Although this mechanism still needs to be tested, it could be conjectured that it is similar to the mechanism used for the cell movement, consisting in the binding of the cytoplasmic domain of PCDH10 with Nap-1, a component of the WAVE complex that regulates actin polymerization [110].

In homozygous mice, KO for *PCDH10* gene, the absence of PCDH10 determines defects in the axon pathways through the ventral telencephalon; in particular, both striatal and thalamocortical projections cannot cross this area [46] showing abnormal wirings.

The crucial role of PCDH10 in synaptic connectivity and circuit formation has been also confirmed during investigations performed on *PCDH10*-heterozygous mice (*PCDH10*^+/−^) where it has been demonstrated that even a partial reduction in functional PCDH10 causes biological alterations at amygdala level that are associated with neurodevelopmental disorders [60]. Amygdala is linked to the social and emotional deficits observed in ASD. The altered expression of PCDH10 at amygdala level has been implicated in the alterations in communication and social behaviors. Recently, it has been observed that the haploinsufficiency of PCDH10 in PCDH10^+/−^ mice reduces the social approach in a gender-sensitive fashion [60] and that this phenotype strongly correlates with electrophysiological and morphological alterations. PCDH10^+/−^ mice show both in vitro and in vivo altered electrophysiological responses. In particular, experiments of voltage sensitive dye imaging showed changes in amygdala circuit synchronization following high stimulation of the synaptic inputs with gamma frequency (40 Hz) [60], supporting the idea that PCDH10 is necessary for the optimal synchrony of the amygdala circuits during the development stages. Analogously, the band responses following in vivo high frequency activity (30–100 Hz, Gamma), that reflect a large-scale brain network functioning fundamental for many cognitive and behavioral functions, result altered [111] like in ASD patients [112] where copy number variation at the *PCDH10* locus has been found [113,114].

Moreover, in the same animals, neuronal morphological alterations have been reported. An increase in the density of the dendritic spines [60], that is an index for the number of excitatory synapses, confirms that PCDH10 is implicated in the developmental synapse elimination. In *PCDH10*^+/−^ mice, the loss of the ability to refine or to eliminate excitatory synapses is perhaps responsible for the overconnectivity/hyperexcitability observed in the amygdala and contributes to the changes in the power of the gamma band activity and the social behavior deficits.

The increase in dendritic spines and in immature spine number occurs also in a genetic model of ASD, the FXS [115], suggesting that such disorder may result from a deficit in synapse elimination involving PCDH10. Fragile X Syndrome is caused by transcriptional silencing or loss-of-function mutations in the *FMRP* gene, which encodes for the Fragile X mental retardation protein (FMRP) that is required for the activity-dependent synapse elimination by PCDH10, triggered following the activation of the transcription factor myocyte enhancer factor 2 (mef2) [116]. It has been demonstrated that both FMRP and mef2 cooperatively regulate the expression of PCDH10 [117] and its binding to PSD-95, a post-synaptic scaffold protein that has a role in anchoring synaptic proteins. In the synapse elimination process, PSD-95 ubiquitinylated binds PCDH10, which links it to the proteasome for degradation. Noteworthy, PSD-95 interacts with GluN2A/2B, two subunits of the NMDA receptor, regulating the development, localization and signaling of this kind of receptor. Thus, the alteration in PCDH10 may indirectly determine the changes in the levels of NMDA receptor subunits, as observed by Schoch and colleagues [60], which would concur with the aberrant oscillatory activity measured in ASD.

### 4.3. Other Protocadherins

#### 4.3.1. PCDH9

PCDH9 is a member of the PCDHδ1 sub-group. It is implicated in the formation of functional neuronal circuits with a specific and restricted spatiotemporal expression pattern. It has been found in different regions of the CNS such as olfactory bulb, hippocampus, caudate putamen and cerebral cortex, since the early developmental stages. However, its expression levels are not constant for the whole life but decrease in adulthood [118,119].

Studies on *PCDH9* KO mice give insight in the possible roles of this protein in neurodevelopmental disorders including ASD (Table 2). The absence of PCDH9, in fact, strongly influences the cortical thickness. This anatomical abnormality is due to an altered synapse development that causes an increased cell density reducing the processing of sensory information relevant for social adaptation [119]. Behavioral analysis of these mice also shows that social long-term memory is impaired [119], clearly suggesting that PCDH9 may participate in the clinical features of ASD, likely through a reduction in the thickness of the cortical areas.

Interestingly, PCDH9 has recently been identified as a risk gene for depressive disorder [120], often occurring in individuals with neurodevelopmental disorders. The analysis of post-mortem brains has shown a decrease in PCDH9 transcript levels in cortex and hippocampus, suggesting that this protein is a key regulator of circuit formation and functioning.

#### 4.3.2. PCDH8

PCDH8 belongs to the δ2 sub-group of PCDHs. It is localized at synapses and neuronal somas of hippocampus, amygdala, and anterior thalamic nuclei of the limbic system where, in addition to the homophilic adhesion activity, modulates synaptic plasticity phenomena that are essential to the process of learning and memory [27] and impaired in neurodevelopmental disorders such as ASD (Table 2).

PCDH8 seems to be implicated in the homeostatic control of the neural complexity. If on one side the disruption of the homophilic interactions mediated by PCDH8, using blocking antibodies, causes a reduction of the amplitude of the excitatory post-synaptic potentials (EPSPs) and a loss of long-term potentiation (LTP) in hippocampal slices [27], on the other side the absence of PCDH8 in hippocampal cultures from *PCDH8* KO mice determines an increase in the dendritic spine density [30]. Although these observations appear to be contradictory, it can be speculated that the control of the spine number by PCDH8 is activated following a synaptic plasticity event and is essential to avoid an excessive neuronal complexity. The mechanism of control of the spine density seems to be dependent on the ability of PCDH8 to interact with *N*-cadherin and promote its endocytosis via the activation of intracellular signaling involving TAO2β and p38MAPK [30]. It is possible that the loss of spines observed in epilepsy, may be explained through this mechanism based on the shedding of *N*-cadherin [121] dependent on PCDH8 activity. Similarly, based on the fact that PCDH8 is an ASD-linked gene [122], it could be speculated that the impairments in synaptic plasticity, which is a causative factor underlying ASD pathology, can be dependent on alterations in PCDH8 functions.

#### 4.3.3. PCDH17

PCDH17 is one of the PCDHδ2 sub-group proteins that plays an important role in cell migration and in axon outgrowth and arborization. Its function has been highlighted in investigations on amygdala neurons where the absence of PCDH17 determines the impossibility of axons to grow normally and their tendency to misroute during the extension [47]. In fact, through the interaction with actin polymerization regulators, via its cytoplasmic domain, PCDH17 creates a kind of machinery at cell-cell contact sites that sustains cell motility. This process allows the migration of growth cones that contact with other axons supporting their collective extension as well as the fascicle formation, which is crucial for a proper neuronal wiring [47].

PCDH17 is predominantly expressed in prefrontal cortex, thalamus, striatum, anterior cingulate and amygdala [60,122,123]. In particular, the enrichment along the amygdala neurons and the basal ganglia synapses has made PCDH17 particularly interesting for its potential implications in mood disorders (Table 2).

In amygdala, the enhanced expression of PCDH17 has been associated to an increased risk to develop major mood disorders, including bipolar disorder, as observed in postmortem brains of patients with higher PCDH17 mRNA levels [124]. The analysis of PCDH17 in primary amygdala cultures has revealed a decrease in the spine density and an aberrant dendritic morphology [124].

The study of PCDH17 in the basal ganglia circuits has highlighted that *PCDH17* KO mice show an antidepressant-like phenotype when challenged with a battery of behavioral tests aimed at evaluating cognition, anxiety, and depression [61], suggesting that this protein could play an important role not only in disease states conceptualized as neurodevelopmental disorders, such as the bipolar disorder, but also in depression that is a comorbid condition in people with neurodevelopmental disorders. Accordingly, single-nucleotide polymorphisms (SNPs) in PCDH17 gene were found associated with mood disorders [124]. Interestingly, in *PCDH17* KO animals, PCDH17 has also been found implicated in the control of vesicle assembly at the basal ganglia circuits where its absence seems to induce an increased number of docked and total synaptic vesicles at presynaptic terminals [61]. Accordingly, when overexpressed in primary neuronal cultures, PCDH17 determines an enhancement in the mobility of synaptic vesicle clusters along the axons [61], suggesting that it inhibits the accumulation of synaptic vesicles.

Collectively, these studies not only reveal that alterations in the levels of PCDH17 cause changes in dendritic spine morphology and abnormal synapse development, which underlie mood disorders, but also that the levels of PCDH17 are crucial for the risk to develop a disease state. Moreover, considering the widespread distribution of PCDH17, it is possible that the alteration in the levels of this protein occurs within an extended brain network that includes not only basal ganglia and amygdala, but also other areas that play an essential role in the control of emotions and social behaviors including cortex, hypothalamus and mesolimbic district.

#### 4.3.4. PCDH15

PCDH15, a member of the PCDHε, is implicated in the regulation of neuronal projections and in synaptic connectivity. It is expressed in brain, kidney, sensory epithelia of the cochlea, and various epithelia during embryogenesis [125].

Deeply investigated for its role in the maintenance of the normal retinal and cochlear function and for its involvement in causing hearing loss and Usher Syndrome Type IF (USH1F) in the presence of mutations in *PCDH15* gene [126], PCDH15 has recently gained attention for its potential implication also in neurodevelopmental disorders (Table 2).

Polymorphisms in *PCDH15* gene have been found in association with schizophrenia [127] and clinically significant copy number variations (CNVs) in *PCDH15* locus have been identified as risk factor for bipolar disorder, ASD, and schizophrenia [128,129,130,131,132,133].

The possible contribution of PCDH15 mutations in causing neurodevelopmental impairment through synaptic alterations has been substantiated in a recent study performed using iPSC lines derived from bipolar patients carrying a CNV in *PCDH15* locus. Exonic deletions in *PCDH15* gene determine a reduction in dendrite length and in the number of both glutamatergic and GABAergic synapses [133]. These data are consistent with studies performed on postmortem human brains of bipolar patients where reduced spine density and dendrite length were observed [134]. Collectively, these observations further confirm the importance of PCDH15 for normal neuronal functions and strongly support the hypothesis that PCDH15 plays a pathogenetic role in neurodevelopmental disorders.

## 5. Conclusions

The proper brain function is critically dependent on the underlying neural architecture and connectivity. The ability of cells to recognize their partners through cell surface receptors, expressed in a given moment, is essential for the formation of a functional neuronal circuitry during development. PCDHs are increasingly recognized as molecules playing important roles in cell adhesion. However, in the last two decades, the research has highlighted that these molecules, in addition to a mechanical role, have other biological functions highly relevant for the neural circuitry development. The expression of PCDHs is tightly regulated during development, and each tissue or cell type shows a characteristic pattern of PCDH expression. This differential expression in the developing nervous system has revealed that PCDHs are implicated in every step of the synapse maturation and circuit formation. This idea has been supported by their involvement in a range of neurodevelopmental disorders and disease states conceptualized as neurodevelopmental disorders (Table 2). Downregulation of PCDHs or their functional alterations have been observed in many human neurodevelopmental diseases. Loss of function in PCDH19, responsible for Developmental and Epileptic Encephalopathy 9, has been reported. Functional alterations determining abnormal synaptic connectivity or plasticity have been found in PCDH-10, -9, and -8, three ASD-linked genes. Changes in PCDH17 and PCDH15 have been implicated in mood disorder onset. These observations clearly demonstrate the relevance of these molecules from a clinical point of view, although the role of many PCDHs in disease etiology still needs to be clarified.

Many advancements in defining PCDH/neurodevelopment relationship have been done. However, the understanding of the mechanisms regulating the activity of PCDHs is still at the beginning due to the high number of molecules expressed, the complexity of their interactions, and the fact that the same PCDHs can mediate opposite functions in different brain districts and at different times. Either the introduction of genetic mutations or the deletion of given PCDHs in animal models have strongly helped to elucidate the physiological significance of these molecules in cell connectivity, synapse maturation, and circuit formation. At the same time, the use of genome-wide association (GWA) studies as an approach to dissect complex human diseases has identified several genes of the PCDH family as possible candidates for neurodevelopmental disorders. However, the limited statistical power has made sure that only few candidate genes of the PCDH family have reached the conventional significance thresholds and have been confirmed as implicated in specific pathological states. Considering the high variety of existing PCDHs, an improvement in GWA analyses will likely uncover further links between PCDHs and neurodevelopmental disorders, boosting the research in this field. Therefore, several important challenges remain to elucidate the PCDHs associated with the neurodevelopmental processes and the molecular mechanisms by which PCDH dysfunctions impact on brain development. This understanding will be critical for the development of effective therapies for these complex conditions.

## Figures and Tables

**Figure 1 cells-09-02711-f001:**
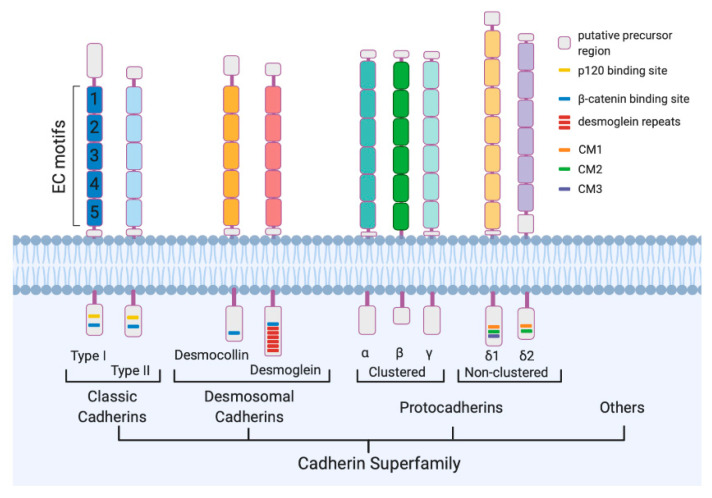
Cadherin superfamily. Schematic drawing of representative members of Classic Cadherins, Desmosomal Cadherins, and protocadherins. All the proteins share “Cadherin motifs” or extracellular cadherin (EC) motifs at their extracellular domain. The number of EC motifs varies from one subfamily to another. The cytoplasmic sequences are conserved only among the members of each family or subfamily.

**Figure 2 cells-09-02711-f002:**
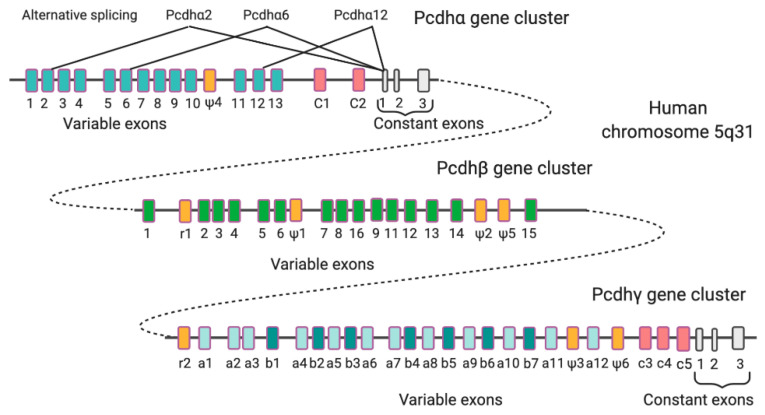
Genomic organization of human *PCDH* gene clusters. Shown are *PCDHα*, *PCDHβ*, and *PCDHγ* gene clusters. Each gene family contains multiple tandem variable exons indicated in a different color; only *PCDHα* and *PCDHγ* share common constant region exons (grey) that form the cytoplasmic tail. In all the three gene clusters are present relic (r) or pseudogene (ψ) variable region sequences (orange). The generation of PCDHa2, PCDHa6, and PCDHa12 through alternative splicing is indicated as an example of exon selection that produces various isoforms.

**Figure 3 cells-09-02711-f003:**
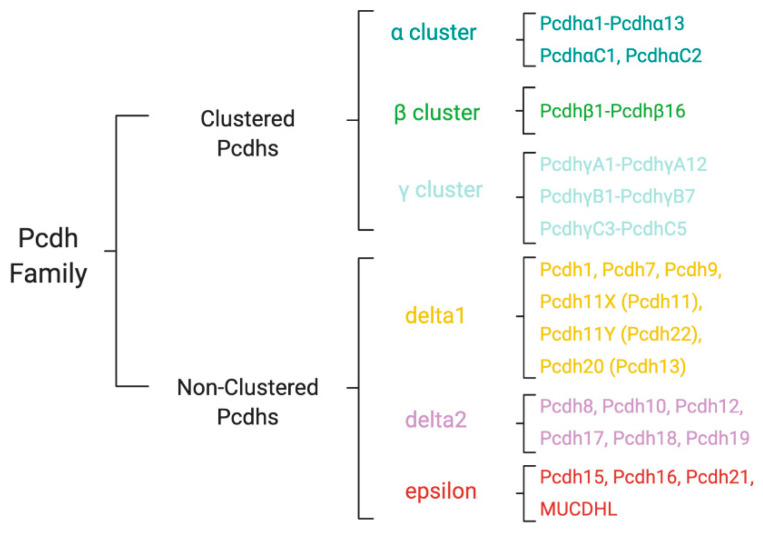
Classification of human protocadherin (PCDH) family. The members of PCDH family are organized in Clustered (alpha, beta, and gamma subfamilies) and Non-Clustered (delta1 and delta2) PCDHs. The Non-Clustered PCDH subfamily also includes PCDHs of the epsilon group.

**Figure 4 cells-09-02711-f004:**
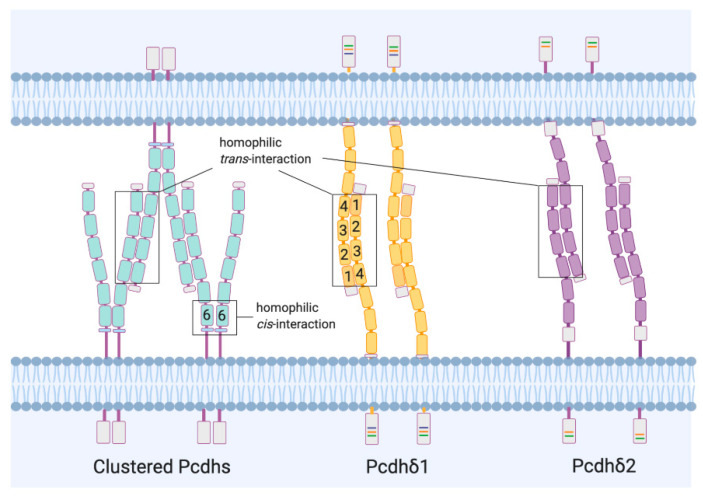
PCDH-mediated cell interaction. Clustered and Non-clustered PCDH isoforms, from different neurites of two cells, recognize each other in an anti-parallel fashion through strict homophilic *trans*-interactions of the EC1-EC4 domains, to mediate the adhesion between cells. In Clustered PCDH isoforms it is also possible to observe promiscuous *cis*-interactions between EC6 domains.

**Figure 5 cells-09-02711-f005:**
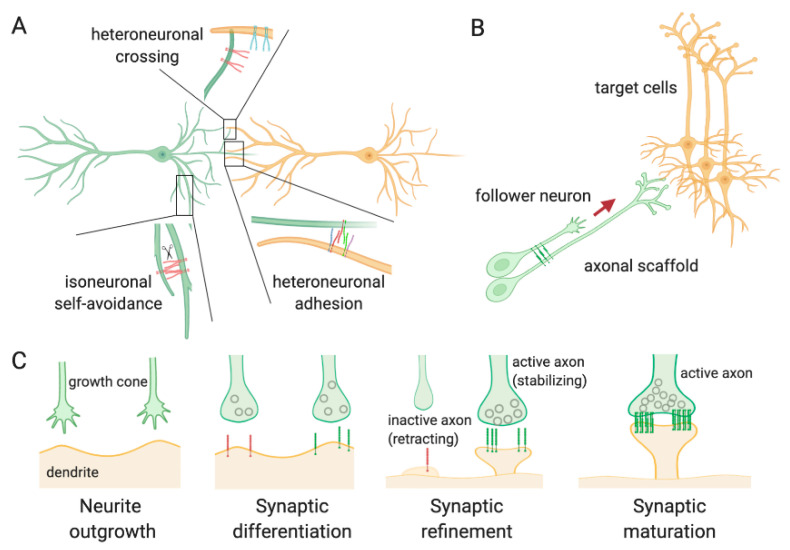
Schematic overview of PCDH functions in neuronal development and circuit formation. (**A**) PCDH-mediated synapse self-recognition and self-avoidance. The expression of identical combinatorial profiles of PCDHs on the same neuron determines repulsion between sister neurites (isoneuronal self-avoidance) following PCDH cleavage and loss of interaction ability. Conversely, when matching PCDHs are expressed on neurites from different cells, the interactions are favored (heteroneuronal adhesion). Lastly, when two neurites from different neurons express distinct combinations of PCDHs, the assembly is interrupted by the mismatched isoforms, and the neurites cross each other (heteroneuronal crossing); (**B**) PCDH regulation of axon-axon interactions through homo- or heterophilic binding between isoforms. The interaction between the growing axon (follower) with a preexisting axon tract (scaffold) via PCDHs mediates the extension of the axon in the appropriate direction, favoring the interaction with other cells; (**C**) Different stages of synapse formation. At an early stage, the dendritic spines are elongated to seek the synaptic partner (neurite outgrowth) and assemble a synapse (synaptic differentiation). The recruitment of PCDHs stabilizes the contacts or determines the retraction of the axon (synaptic refinement), according to the isoforms of PCDHs expressed and to the synaptic activity/plasticity. PCDH complexes promote the expansion of the dendritic spine head and the maturation of the synapse.

**Table 1 cells-09-02711-t001:** Summary of PCDH mRNA expression in human tissues. Shown are the major PCDHs implicated in neurodevelopmental disorders. Level of expression is indicated: +++ strong; ++ moderate; + weak; - negative.

		Clustered PCDHs			Non-Clustered PCDHs					
Major Tissues		α	β	γ	PCDHδ1	PCDHδ2				PCDHε
					9	8	10	17	19	15
Nervous	Whole Brain	+++	+++	+++	+++	+++	+++	+++	+++	+++
	Cortex	+	+++	+++	+++	+++	+++	+++	+++	+++
	Hippocampus	++	+++	+++	+++	+++	+++	+++	+++	+++
	Cerebellum	++	+++	+++	+++	++	++	+++	++	++
	Retina	+++	-	-	-	-	-	-	-	++
	Spinal Cord	+	+++	+++	+++	++	++	+++	++	+++
	Tibial Nerve	++	+++	+++	+++	++	++	++	++	+
Muscle	Heart	+	++	+++	++	+	+	++	++	++
	Artery	++	+++	+++	++	+	++	++	++	+
	Smooth Muscle	-	-	-	-	-	-	-	-	-
	Skeletal Muscle	+	++	++	++	+	+	++	++	+
Internal	Small Intestine	+	++	++	+	+	++	++	++	++
	Colon	+	+++	+++	++	+	++	+++	+++	++
	Adipocyte	++	+++	+++	++	+	++	++	+++	+
	Kidney	++	++	++	++	+	+++	++	+	++
	Liver	+	++	++	+	+	+	++	+	+
	Lung	+	+++	+++	+	+	++	+++	+	+++
	Spleen	+	++	++	+	+	+	+++	+	++
	Stomach	+	++	++	++	+	++	++	++	+
	Esophagus	+	++	+++	++	+	+++	++	++	+
	Bladder	+	+++	+++	++	+	+++	++	++	+

**Table 2 cells-09-02711-t002:** PCDHs associated with neurodevelopmental disorders. Shown are the members of the PCDH family, their physiological functions in neuronal circuitry development, the disorders associated with their alterations and the main brain areas involved.

Name	Circuitry Related Functions	Related Diseases	Brain Region Involved
PCDH19	Neural association and proliferation	Epilepsy, intellectual disability, ASD	Amygdala, hippocampus, cortex, hypothalamus
PCDH10	Synapse elimination, axon guidance and formation	ASD	Basal ganglia, amygdala
PCDH9	Neurite localization	ASD, schizophrenia, depression	Cortex, hippocampus
PCDH8	Synapse elimination	ASD	Hippocampus
PCDH17	Axon growth and fasciculation, synapse development, synaptic vesicles	Bipolar disorder, schizophrenia, depression	Amygdala, basal ganglia
PCDH15	Neuronal projections and connectivity	Schizophrenia, bipolar disorder, ASD	Cortex

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
