# Peer review of "Right Place at the Right Time: How Changes in Protocadherins Affect Synaptic Connections Contributing to the Etiology of Neurodevelopmental Disorders"

_cells, 2020, doi:10.3390/cells9122711_

Round 1

Reviewer 1 Report

This is a well-written and straightforward review that describes the roles of protocadherins in various neurodevelopmental disorders and synaptic connectivity. The review discusses the types of protocadherins and, for each of them, outlines their specific role.

Abstract, line 15: change ‘take part to neurodevelopmental process’ to ‘take part in the neurodevelopmental process’

Line 17: Change ‘because are involved’ to ‘because they are involved’

Line 61: ‘and-catenin’. There are extra symbols in front of catenin.

Line 73: ‘in clusters’ should it be ‘in clustered

Line 131: are you suggesting that PCDHs are expressed on axons and, therefore, they affect to target areas or how do projections affect the circuitry?

Line 139: change ‘take part to the process’ to ‘take part in the process’

Line 348: explain the ‘cellular interference hypothesis’

Line 382: spell out ASD at its first use

Line 395: change ‘heterozigous’ to ‘heterozygous’

Author Response

This is a well-written and straightforward review that describes the roles of protocadherins in various neurodevelopmental disorders and synaptic connectivity. The review discusses the types of protocadherins and, for each of them, outlines their specific role.

We thank the reviewer for the positive comments about our manuscript.

*Abstract, line 15: change ‘take part to neurodevelopmental process’ to ‘take part in the neurodevelopmental process’

Thank you for noticing this imprecision. It has been corrected.

*Line 17: Change ‘because are involved’ to ‘because they are involved’

We made this change.

*Line 61: ‘andb-catenin’. There are extra symbols in front of catenin.

Thanks for catching the typo. The extra space before ‘b-catenin’ has been removed.

*Line 73: ‘in clusters’ should it be ‘in clustered

The mistake has been amended.

*Line 131: are you suggesting that PCDHs are expressed on axons and, therefore, they affect to target areas or how do projections affect the circuitry?

We appreciate this question that allows us to further discuss the functional roles of PCDHs. Both clustered and non-clustered PCDHs are localized in dendrites, axons and proximally to or within synapses. During neuronal wiring, axons establish a framework that is dependent on a series of guidance events during neurodevelopment that also include the expression of combination of PCDHs. PCDHs at axonal sites not only make possible interactions between neighboring cells influencing their spatial organization but, as discussed in the section 3.2, they are also involved in axon outgrowth that is crucial to ensure a proper synaptic connectivity. Thus, to address the reviewer’s question, are the axonal projections to determine the impact of a given cell on other neurons and on the whole circuitry. However, it cannot be ruled out that different cells in contact through their axons influence each other in a way depending on the kind of PCDH expressed of the cell surface.

*Line 139: change ‘take part to the process’ to ‘take part in the process’

The change at line 119 has been performed as suggested.

*Line 348: explain the ‘cellular interference hypothesis’

We appreciate the suggestion. The revised version of the manuscript contains an explanation of the ‘cellular interference hypothesis’ at line 257.

*Line 382: spell out ASD at its first use

The acronym ASD was introduced and spelled out at line 228.

*Line 395: change ‘heterozigous’ to ‘heterozygous’

A correction of the word ‘heterozygous’, misspelled at line 358, has been done.

Reviewer 2 Report

Line 61: p120 ctn: p120 catenin

Line 61: and 00 β-catenin: ??

Line 61: 0α-catenin: ??

Line 68: 5 EC repeats: EC full name

Line 83-84: the reasons why you illustrated human and rat? Are these important?

The authors introduced several PCDHs expressed in the limbic system and amygdala in this paper, and a table regarding different PCDHs related diseases and possible mechanisms may help readers to have a general concept for the same and different functions for PCDHs in these areas.

Line 512: Here is the first time this paper described the polymorphisms in PCDH15 gene, but why the authors did not mention polymorphisms of other PCDHs ? for example

PCDH17 polymorphisms are related to depression https://www.nature.com/articles/mp2016231

PCDH8 polymorphisms are related to schizophrenia

https://pubmed.ncbi.nlm.nih.gov/12884975/

Why the authors ignored these reports?

Conclusion is relatively weak.

Author Response

Comments and Suggestions for Authors

*Line 61: p120 ctn: p120 catenin

We thank the reviewer for highlighting the inaccuracy. The abbreviation ‘ctn’ has been changed with the full word ‘catenin’.

*Line 61: and 00 β-catenin: ??

Thanks for catching the error. The extra space before ‘b-catenin’ has been removed.

*Line 61: 0α-catenin: ??

Correction has been made, thank you.

*Line 68: 5 EC repeats: EC full name

The acronym EC was introduced and spelled out at line 58.

*Line 83-84: the reasons why you illustrated human and rat? Are these important?

Although interesting to report the genome organization of PCDH gene clusters of rat or other species, we preferred to show in figure 2 only the human gene. Moreover, PCDH genes in human, mouse, rat and zebrafish are very conserved (Wu Q et al., 2001 Genom Res 11, 389-404; Noonan JP et al., 2004 Genom Res 14, 354-366). Our goal was only to give to the reader an idea of the cluster organization.

*The authors introduced several PCDHs expressed in the limbic system and amygdala in this paper, and a table regarding different PCDHs related diseases and possible mechanisms may help readers to have a general concept for the same and different functions for PCDHs in these areas.

We are grateful with reviewer’s suggestion. A table has been introduced in the review and named ‘Table 2’.

*Line 512: Here is the first time this paper described the polymorphisms in PCDH15 gene, but why the authors did not mention polymorphisms of other PCDHs ? for example

PCDH17 polymorphisms are related to depression https://www.nature.com/articles/mp2016231

PCDH8 polymorphisms are related to schizophrenia

https://pubmed.ncbi.nlm.nih.gov/12884975/

Why the authors ignored these reports?

We thank the reviewer for this criticism. We are aware that some of the elegant studies that the reviewer mentioned, and that we already cited in the review (Chang H et al., 2018, Mol Psychiatry 23, 400-412), report also in other PCDHs the existence of polymorphisms related to disease states conceptualized as neurodevelopmental disorders. In describing both PCDH17 and PCDH8 we focused on other aspects without putting the accent on the polymorphisms. While we totally agree with the reviewer’s suggestion to introduce PCDH17 polymorphisms in our report, in the modified version we did not elaborate on polymorphisms of PCDH8 gene. The study about a possible contribution of PCDH8 polymorphisms to schizophrenia susceptibility, performed by Bray NJ and colleagues (2002, Genes Brain Behav 1, 187-191) and suggested by the reviewer, reveals no strong association with the disease. For such reason, we do not consider it so relevant to be discussed.

*Conclusion is relatively weak.

Changes in the Conclusions have been made.